# Prevalence and Socio-Demographic Correlates of Dental Anxiety among a Group of Adult Patients Attending Dental Outpatient Clinics: A Study from UAE

**DOI:** 10.3390/ijerph20126118

**Published:** 2023-06-13

**Authors:** Al Batool Omer Alansaari, Abdelrahman Tawfik, Mohamed A. Jaber, Amar Hassan Khamis, Essra Mohamed Elameen

**Affiliations:** 1Hamdan Bin Mohammed College of Dental Medicine, MBRU, Dubai P.O. Box 505055, United Arab Emiratesabdel.tawfik@mbru.ac.ae (A.T.); amar.hassan@mbru.ac.ae (A.H.K.); 2Clinical Sciences Department, College of Dentistry, Ajman University, Ajman P.O. Box 346, United Arab Emirates; 3Clinical Sciences Department, College of Medicine, Ajman University, Ajman P.O. Box 346, United Arab Emirates; essrajaber@gmail.com

**Keywords:** anxiety scale, dental phobia, treatment, distraction, fear

## Abstract

Objectives: The aims of this paper were twofold: first, to evaluate dental anxiety levels among patients undergoing oral surgery procedures; second, to assess how dental anxiety and fear are connected to age, gender, educational background, past traumatic experiences, and the frequency of dental appointments. Methods: A cross-sectional Likert-scale questionnaire survey was conducted to collect quantitative data from 206 patients at the Oral Surgery Clinics at Dubai Dental Clinics, Dubai, United Arab Emirates. The reliability and validity of the questionnaire were tested using Cronbach’s alpha. The normality of the MDAS score was tested using the Kolmogorov–Smirnov test. The chi square and Kruskal–Wallis tests were used to determine the association between categorical variables. Descriptive statistics were used to describe continuous and categorical variables. The statistical significance was set at *p*-value ≤ 0.05. Results: The evaluation of the degree of dental anxiety among patients who visited the Dubai Dental clinics revealed that there was a considerably high level of moderate or high anxiety (72.3%). Tooth extraction and dental surgery procedures (95%), followed by local anesthetic injection in the gingiva (85%) and teeth drilling (70%), were the primary causes of anxiety, whereas scaling and polishing resulted in the lowest degree of anxiety (35%). There was no substantial difference in dental anxiety between male and female patients or among patients with different marital statuses. A total of 70% of patients preferred the tell-show-do method, whereas 65% chose communication strategies to reduce dental anxiety. Conclusions: The evaluation of the degree of dental anxiety among patients who visited Dubai Dental clinics revealed that there was a considerably high level of anxiety. Tooth extraction and dental surgery procedures, followed by local anesthetic injection and teeth drilling, were the primary causes of anxiety, whereas scaling and polishing resulted in the lowest degree of anxiety. Despite the use of a modified anxiety scale and a large and representative sample of oral surgery patients, more research is necessary to investigate the impact of various factors on dental anxiety.

## 1. Introduction

Anxiety is a negative emotional state that can lead to nervous behavior, such as nail biting or teeth grinding, and is often accompanied by a sense of dread over anticipated events. In contrast to fear, which is a response to a known danger, anxiety arises from an expectation of future threats. The primary cause of dental anxiety is typically previous painful or traumatic dental experiences, which can lead to negative thoughts about dental treatment and an increased sensitivity to pain [1].

It can be defined as a feeling of nervousness or apprehension that is associated with dental treatment and can range from mild discomfort to a more severe form that can interfere with treatment and cause significant distress to the patient [1]. Dental anxiety is a common phenomenon that is experienced by many patients who attend oral surgery clinics and is more prevalent in women than men [2]. Factors that can contribute to dental anxiety include previous negative experiences, fear of pain, and fear of losing control during treatment [3,4]. Patients with dental anxiety are more likely to avoid dental treatment, which can lead to poor oral health outcomes [5]. Additionally, dental anxiety can increase the risk of complications during treatment, as patients may move or flinch during procedures, making it more difficult for the dentist to perform the necessary procedures [6], furthermore, dental anxiety can be associated with significant distress and impaired quality of life [7]. There are several interventions that can be used to manage dental anxiety in patients who attend oral surgery clinics. One approach is to use pharmacological interventions, such as benzodiazepines or nitrous oxide, to reduce anxiety levels [8]. However, these interventions can have side effects and may not be suitable for all patients. Another approach is to use non-pharmacological interventions, such as cognitive behavioral therapy (CBT) or hypnosis, to manage anxiety levels [9]. CBT involves identifying and challenging negative thoughts and beliefs about dental treatment, whereas hypnosis involves inducing a state of deep relaxation to reduce anxiety levels. Both of these interventions have been shown to be effective in reducing dental anxiety [10,11]. In addition to these interventions, there are several other strategies that can be used to manage dental anxiety. These include providing patients with information about the procedure and what to expect, using distraction techniques such as music or television during treatment, and creating a calming environment in the dental office [12]. Additionally, some dentists may use techniques such as positive reinforcement and relaxation training to help patients manage their anxiety levels [13]. The prevalence of clinical dental anxiety ranges from about 4% to over 20% worldwide, with 48% of adults estimated to experience dental anxiety in the UK [8,12]. Similarly, a survey in Canada found that 9.8% of respondents were somewhat afraid of dental visits, whereas 5.5% were very afraid or terrified [13]. In the Gulf region, a study in Saudi Arabia revealed that 48.3% of the population had dental anxiety [14]. However, there is no research available to assess anxiety levels among the UAE population. Hence, the aims of this paper were twofold: first, to evaluate dental anxiety levels among UAE patients undergoing oral surgery procedures; second, to assess how dental anxiety and fear are connected to age, gender, educational background, past traumatic experiences, and the frequency of dental appointments.

## 2. Materials and Methods

### 2.1. Design and Population

A cross-sectional study was conducted at the Oral Surgery Clinics of Hamdan Bin Mohammed College of Dental Medicine, Dubai, United Arab Emirates, to determine the level of dental anxiety among outpatients.

The samples chosen for the study consisted of individuals who had scheduled appointments at the oral surgery department. The selected group encompassed a variety of participants from diverse demographic backgrounds.

### 2.2. Sample Size

According to the paper published by Ulf Berggren and Gunnell Meynert in Dharan-KSA and using Cochrane sample size calculation:$$n=z^{2}\alpha /2=\frac{p(1-p)}{d^{2}}$$

With given data from that paper (*n* = 824, and anxiety was 90.9%) using the width of the 95% CI, the sample size is 126. Due to the population diversity in Dubai, we doubled the sample size and obtained 206, which is above the calculated sample size targeted.

### 2.3. Data Collection

A standardized questionnaire in both the Arabic and English languages were used to collect data. The questionnaire had three parts. The first part collected socio-demographic information. The second part had questions related to how the patient felt about upcoming dental visits, the dentist’s waiting room, having teeth drilled or scaled and polished, local anesthetic injection, and oral surgery procedures. The modified dental anxiety scale (MDAS), a self-reported measure with a five-point Likert scale (1 = not anxious to 5 = extremely anxious), was used in this part to determine the level of dental anxiety. The total score ranged from 5 to 30. The third part of the questionnaire consisted of one question where patients were asked to select the most useful technique from nine options to reduce their dental anxiety. MDAS scores were categorized as low (5–12), moderate (13–18), and high (≥19) dental anxiety.

The reliability and validity of the questionnaire was tested by Cronbach’s alpha in SPSS (Statistical Package for Social Sciences) for Windows version 24 to insure the suitability of the test and the quality and usefulness of the study. Additionally, test–retest reliability was performed to make sure that the data were repeatable and consistent. In terms of inter-rater reliability, only one researcher was involved in collecting the data to prevent any variations in the test scores; the involvement was controlled to prevent any bias.

### 2.4. Data Analysis

IBM SPSS version 24 (Chicago) was used to analyze the data. The reliability and validity of the questionnaire were tested using Cronbach’s alpha. The normality of the MDAS score was tested using the Kolmogorov–Smirnov test. The chi square and Kruskal–Wallis tests were used to determine the association between categorical variables. Descriptive statistics were used to describe continuous and categorical variables. Statistical significance was set at *p*-value ≤ 0.05.

### 2.5. Ethical Approval

The study adhered to the principles of the Helsinki Declaration, and ethical approval was obtained from the Research Ethics Review Committee of HBMCDM. All participants gave written consent to participate in the study.

## 3. Results

### 3.1. Socio-Demographical Data

The total number of participants who completed the questionnaire was 206 patients. The demographic data of the participants is shown in Table 1.

The sample was comprised of 53% males and 47% females. The majority of the sample (44%) was aged between 31 and 40 years and 13% of the participants were above 51 years of age. A total of 74% of the patients were married and employed (77%), whereas only 14% were unemployed and 9% were students. A total of 38% of the respondents had a bachelor’s degree, 28% had a diploma degree, 21% had a school degree, and 12% had a master’s degree. More than half of the patients in the sample were an irregular visitor to the dentist (56%), whereas 35% of the patients visited the dentist regularly and only 9% did not visit the dentist at all. A total of 21% of the patients had had a bad dental experience.

### 3.2. Modified Dental Anxiety Scale

Using the modified dental anxiety scale (MDAS), the participants in this study have a moderate level of dental anxiety, with a total score of 15.50 ± 5.4. Among the specific dental procedures, having a tooth drilled and having an extraction or surgical procedure are associated with higher levels of anxiety (fairly anxious to very anxious). Having a local anesthetic injection is also associated with a fairly anxious level of anxiety, whereas the other procedures (going to the dentist for treatment tomorrow, sitting in the dentist waiting area, and having teeth scaled and polished) are associated with a slightly anxious level of anxiety (Table 2).

### 3.3. Group Comparison

Table 3 presents the characteristics, levels, and percentages of dental anxiety among different groups of participants based on their gender, age, nationality, marital status, occupation, and education. The results show that there is a slightly higher proportion of participants with high dental anxiety among females compared with males, although the difference is not statistically significant (*p* = 0.084). Age does not seem to be associated with dental anxiety levels, as there is no significant difference between the four age groups (*p* = 0.866). Nationality and marital status also do not show significant associations with dental anxiety levels (*p* = 0.623 and *p* = 0.631, respectively). However, occupation and education are significantly associated with dental anxiety levels. Participants who are unemployed have a higher proportion of high dental anxiety levels than students and employees (*p* = 0.034). Education level is also significantly associated with dental anxiety levels, with participants who have a higher education level having a higher proportion of high dental anxiety levels (*p* = 0.004). Specifically, participants with a diploma have a lower proportion of high dental anxiety levels compared with those with a bachelor’s or master’s degree.

Table 4 presents the mean ± sd (standard deviation) of anxiety scores among participants categorized by different dental characteristics. The results show that participants who reported having a regular dental checkup had a higher anxiety score (15.49 ± 4.32) than those who had irregular dental checkups (15.19 ± 5.4); however, those with no history of a dental visit reported the highest dental anxiety scores (17.05 ± 8.44). However, the *p*-value of 0.57 indicates that the difference in anxiety scores between these groups is not statistically significant. The participants who reported having good oral hygiene had a slightly lower anxiety score (15.3 ± 4.7) compared with those with moderate (15.55 ± 5.73) and poor (15.94 ± 7.12) oral hygiene. However, the *p*-value of 0.999 indicates that there is no statistically significant difference in anxiety scores between these groups. Likewise, participants who reported having a bad dental experience had a slightly higher anxiety score (16.95 ± 6.97) compared with those who did not report a bad dental experience (*p*-value = 0.11). However, this difference is not statistically significant (*p* < 0.05).

The majority of the patients chose communication strategies 104 (51%), followed by tell-show-do techniques 68 (33%) and relaxation strategies and distraction techniques 62 (30%) as the best techniques that could be used to reduce anxiety. Lastly, the results showed that GA, cognitive behavioral therapy, and seeing a psychologist were on the bottom of the rank, with 15 (7%), 6 (3%), and 6 (3%), respectively (Figure 1).

## 4. Discussion

In this study, the average total score for dental anxiety was found to be 15.50 (SD ± 5.4), which is higher than the scores reported in previous studies conducted in India [15], China [16], Turkey [17], Greece [18], Saudi Arabia [19], Spain [20], and Iran [21], which ranged from 8.7 to 12.3. It is unclear why the current study produced higher scores than those reported in previous studies. One possibility is that all the patients in the current study underwent different oral surgery procedures, which are generally more invasive than routine dental check-ups or restorative work, which could account for the higher dental anxiety scores; likewise, other factors such as sample selection (i.e., general population or patients scheduled for intervention) or ethnic and sociocultural variables may account for the differences between this study and other published reports.

In this study, we used the modified dental anxiety scale (MDAS), which is a questionnaire used to assess the level of dental anxiety in individuals [22]. The participants have either moderate or high level of dental anxiety, with a total score of 15.50 ± 5.4. Among the specific dental procedures, having a tooth drilled and having an extraction or surgical procedure are associated with higher levels of anxiety (fairly anxious to very anxious). Having a local anesthetic injection is also associated with a fairly anxious level of anxiety, whereas the other procedures (going to the dentist for treatment tomorrow, sitting in the dentist waiting area, and having teeth scaled and polished) are associated with a slightly anxious level of anxiety; these results are in agreement with previous studies [16,17,19]. Moreover, in this study, there is no significant relationship between age, nationality, marital status, and dental anxiety level.

Females have a higher percentage of moderate and high dental anxiety levels compared with males, although the difference is not statistically significant (*p* = 0.084). Previous reports indicated that dental anxiety was related to personality and psychological status [23,24]. This finding might be explained on the basis that women have higher levels of neuroticism than men and that anxiety is positively associated with neuroticism [25,26]. However, in the current study, the statistically significant difference between men and women was marginal and the difference in anxiety scores for both genders was minimal. Therefore, it could be inferred that statistical significance might not be necessarily interpreted as a clinical one.

The results of this study suggest that as educational level increases, so do dental anxiety scores, which is consistent with another study [27]. However, other studies [19,21] have found no effect of education level on dental anxiety or have shown that patients with lower levels of education have higher anxiety scores. It is possible that those with more education may be better equipped to cope with anxiety and stress. Additionally, the study found a significant association between dental anxiety and employment status, with students and unemployed patients exhibiting higher anxiety scores than employed patients. Further research is needed to understand why this is the case.

The findings of this review suggest that there may be differences in dental anxiety levels between students and employed individuals, although the direction of these differences is inconsistent across studies. For example, it was reported that students had higher levels of dental anxiety than employed individuals [28,29,30,31]. Likewise, other reports found no significant differences in dental anxiety levels between students and employed individuals [30]; in contrast, another study found that employed individuals had higher levels of dental anxiety than students [29].

The reasons for these differences are not clear but may be related to factors such as age, experience with dental procedures, and socioeconomic status. It is also possible that these differences are influenced by cultural factors, such as attitudes towards dental care and the role of dental health in society.

The results of this study did not find any relationship between dental anxiety scores and self-reported oral hygiene; however, it is possible that this measure may not accurately reflect actual oral hygiene.

Our results suggest that having no history of dental visits may be associated with higher anxiety scores compared with being a regular dental visitor, whereas bad dental experiences may be associated with slightly higher anxiety scores. However, the differences observed are not statistically significant except for the possible association between bad dental experiences and higher anxiety scores, which requires further investigation with a larger sample size. The lack of statistical significance in the other comparisons may indicate that these dental characteristics are not major predictors of anxiety in the study population. These findings are consistent with a previous study that found patients who have visited a dentist before have less anxiety than those who have not and that those who have had a bad dental experience have higher levels of anxiety [32]. In contrast, other reports found no significant association between previous dental visits and dental anxiety [33]. Furthermore, patients who have a scheduled appointment may have more time to anticipate and worry about the procedure, which could potentially increase their anxiety. On the other hand, unscheduled or emergency cases might lead to higher anxiety due to the unexpected nature of the visit. However, the specific influence of scheduled treatment on anxiety levels would depend on various factors, including individual patient characteristics, previous experiences, and overall anxiety levels.

Dental anxiety is a common problem that can be triggered by specific dental procedures. For example, certain dental procedures are associated with higher levels of anxiety than others, including having a tooth drilled, having an extraction or surgical procedure, and having a local anesthetic injection. One of the most anxiety-provoking dental procedures is tooth drilling. The sound and vibration of the drill can be uncomfortable for patients, and the fear of pain during the procedure can be a significant source of anxiety [23]. Additionally, patients may feel a loss of control during the procedure, as they are unable to see what is happening inside their mouth.

Likewise, having a tooth extraction or surgical procedure can also be a source of anxiety for patients. The fear of pain during the procedure, as well as the potential for bleeding and swelling afterwards, can be intimidating [23]. Patients may also worry about the need for general anesthesia or other medications that may have side effects. Furthermore, a local anesthetic injection is often necessary for many dental procedures, including tooth drilling and extraction. However, the injection itself can be a significant source of anxiety for patients. The fear of needles and injections is common, and patients may worry about the pain and discomfort associated with the injection [32].

Although going to the dentist for treatment tomorrow may seem like a routine procedure, it can still be a source of anxiety for some patients. The fear of the unknown, including what procedures may be necessary and how much they will cost, can be stressful [23]. Patients may also worry about taking time off work or school for the appointment.

Moreover, sitting in the dentist waiting area may be associated with a slightly anxious level of anxiety because patients may feel exposed and vulnerable while waiting for their appointment. They may worry about other patients overhearing their conversation with the receptionist or about running into someone they know [23].

Having teeth scaled and polished is generally a less anxiety-provoking dental procedure. However, patients may still feel slightly anxious due to the discomfort associated with the scaling and polishing process, as well as the fear of being judged for their dental hygiene habits [23].

Dental professionals should be aware of these potential sources of anxiety and take steps to help alleviate patients’ fears.

Dental anxiety affects people of different ages, genders, and backgrounds, and the level of dental anxiety may be affected by various factors, including occupation and level of education. Several studies have shown that people in certain occupations may experience higher levels of dental anxiety than others. For example, healthcare workers, particularly those who work in hospitals, may have higher levels of dental anxiety due to their knowledge of the potential risks associated with dental procedures [32]. Similarly, individuals who work in public-facing roles, such as salespeople and teachers, may experience higher levels of dental anxiety due to concerns about their appearance and the impact of dental procedures on their professional image [29]. Level of education has also been identified as a factor that may influence dental anxiety levels. Several studies have found that people with lower levels of education tend to have higher levels of dental anxiety than those with higher levels of education [4,23]. This may be due to the fact that people with lower levels of education may have less knowledge about dental procedures, making them more fearful of the unknown.

Dental anxiety can have a significant impact on patients’ willingness to seek dental treatment. However, the impact of dental anxiety is not limited to the psychological level; it can also have biological implications for patients undergoing dental treatment. Research has shown that dental anxiety can lead to a range of biological responses that can impact dental treatment. For example, dental anxiety has been found to be associated with increased levels of stress hormones such as cortisol, adrenaline, and noradrenaline [4,33]. These stress hormones can have a range of effects on the body, including increasing heart rate and blood pressure and reducing blood flow to the gums, which can affect the healing process after dental procedures.

In addition, dental anxiety has been linked to a heightened pain response during dental treatment. This may be due to the release of stress hormones that can amplify pain perception [4,34]. The fear of pain can also lead patients to avoid seeking dental treatment altogether, which can lead to more severe dental problems in the long run.

Furthermore, dental anxiety can lead to increased muscle tension in the jaw, neck, and shoulders, which can make it difficult for the dentist to perform dental procedures [23]. This can also lead to increased discomfort and pain during dental treatment, which can further exacerbate patients’ anxiety.

Effective management of dental anxiety can be achieved through various means such as providing good dental health education, ensuring regular dental visits, fostering a positive patient–dentist relationship, and maintaining appropriate communication with patients [35]. It is important to approach patients with dental anxiety in a gentle, supportive, professional, sympathetic, quiet, and considerate manner, especially during their first visit, to avoid exacerbating their anxiety and aversion to dental care [36].

Various techniques can be used to manage dental anxiety, including seeking the help of a psychologist or psychiatrist, implementing effective communication strategies, using distraction techniques such as music and movies, adopting relaxation strategies such as boxed breathing and progressive muscles relaxation, applying cognitive behavioral therapy, using nitrous oxide or conscious sedation through pharmacological drugs such as midazolam or general anesthesia [12,13,37].

Since dental anxiety can arise from various factors, it is important to assess anxiety levels before dental treatment to avoid complications and to ensure effective treatment. Empathy from the dentist, appropriate behavior, and the use of sedation and hypnosis can all contribute to a comfortable dental experience. Anxiolytic drugs can also be used to achieve full conscious sedation, with general anesthesia reserved for specific cases.

Studies of this kind will unavoidably face certain limitations. Despite careful screening to exclude patients with psychological disorders that could impact anxiety assessment, there is a possibility that some patients may have been overlooked, as the reliability of the study was reliant on patient self-reporting. Another recognized limitation is the cross-sectional nature of the survey, which cannot establish causality. Furthermore, the study employed a small sample size and a self-administered questionnaire, which may be subject to bias, as patients may over or underestimate their responses.

## 5. Conclusions

The evaluation of the degree of dental anxiety among patients who visited Dubai Dental clinics revealed that there was a considerably high level of anxiety. Tooth extraction and dental surgery procedures, followed by local anesthetic injection and teeth drilling, were the primary causes of anxiety, whereas scaling and polishing resulted in the lowest degree of anxiety. Despite the use of a modified anxiety scale and a large and representative sample of oral surgery patients, more research is necessary to investigate the impact of various factors on dental anxiety.

## Figures and Tables

**Figure 1 ijerph-20-06118-f001:**
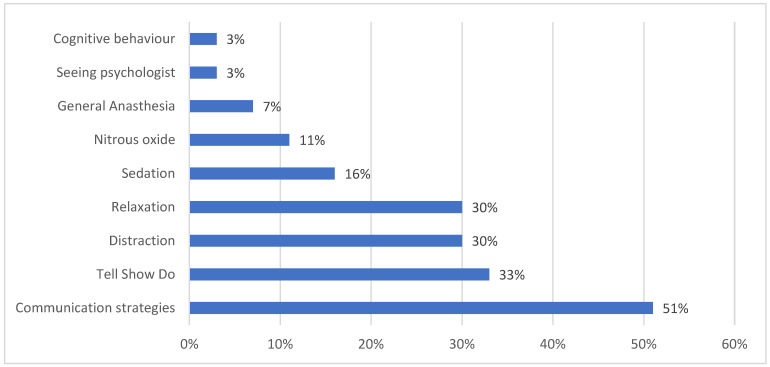
Best technique that can be used to reduce the dental anxiety.

**Table 1 ijerph-20-06118-t001:** Socio-Demographical data statistical distribution.

Characteristics	No	%
Gender		
• Male	110	53.4
• Female	96	46.6
• Total		100
Age		
• <20	10	5
• 21–30	38	18.4
• 31–40	91	44.2
• 41–50	40	19.4
• >50	27	13.1
Marital status		
• Single	54	26
• Married	152	74
Nationality		
• UAE	33	16
• Expatriate	173	84
Employment status		
• Unemployed	29	14
• Employed	159	77
• Students	18	9
Education level		
• School	44	21.3
• Diploma	58	28.1
• Bachelor	79	38.3
• Postgraduate	25	12.1
Anxiety score		
• Low	57	28
• Moderate	99	48
• High	50	24
Dental history		
• No history of dental visit	19	9.2
• Irregular dental checkup	116	56.3
• Regular dental checks up	71	34.5
Bad dental experience		
• Yes	43	21
• No	163	79

**Table 2 ijerph-20-06118-t002:** Mean Modified Dental Anxiety Scale (MDAS).

MDAS Items	Mean ± sd	Classification
Going to the dentist for treatment tomorrow	2.12 ± 1.04	Slightly Anxious
Sitting in the dentist waiting area	2.21 ± 1.03	Slightly Anxious
Having a tooth drilled	2.70 ± 1.19	Fairly Anxious
Having teeth scaled and polished	1.92 ± 1.04	Slightly Anxious
Having local anesthetic injection	3.05 ± 1.25	Fairly Anxious
Having extraction or surgical procedure	3.50 ± 1.25	Fairly Anxious to very anxious
Total score	15.50 ± 5.4	Moderately Anxious

**Table 3 ijerph-20-06118-t003:** Comparison of Dental Anxiety score by socio-demographic characteristics.

Characteristics	Levels	Low Dental Anxiety	Moderate Anxiety	High Dental Anxiety	*p*-Value
Gender	Male	37 (33.6)	51 (46.4%)	22 (20.0%)	0.084
	Female	20 (20.8%)	48 (50%)	28 (29.2%)	
Age	<30	13 (27.1%)	20 (41.7%)	15 (31.3%)	0.866
	31–40	24 (26.4%)	47 (51.6%)	20 (22.0%)	
	41–50	12 (30.0%)	18 (45.0%)	10 (25.0%)	
	≥51	8 (29.6%)	14 (51.9%)	5 (18.5%)	
Nationality	Local	9 (27.3%)	18 (54.5%)	6 (18.2%)	0.623
	Expatriate	48 (27.7%)	81 (46.8%)	44 (25.4%)	
Marital Status	Single	17 (31.5%)	23 (42.6%)	14 (25.9%)	0.631
	Married	40 (26.3%)	76 (50.0%)	36 (23.7%)	
Occupation	Unemployed	4 (13.8%)	14 (48.3%)	11 (37.9%)	0.034 *
	Student	4 (22.2%)	6 (33.3%)	8 (44.4%)	
	Employee	49 (30.8%)	79 (49.7%)	31 (19.5%)	
Education	School	19 (43.2%)	16 (36.4%)	9 (20.5%)	0.004 *
	Diploma	12 (20.7%)	39 (67.2%)	7 (12.1%)	
	Bachelor	20 (25.3%)	35 (44.3%)	24 (30.4%)	
	Master	6 (24.0%)	9 (36.0%)	10 (40.0%)	

* significant.

**Table 4 ijerph-20-06118-t004:** Comparison of the score of Anxiety by dental characteristics.

Variables		No (%)	Mean ± sd	*p*-Value
Dental history	No history of dental visit	19 (9.2)	17.05 ± 8.44	0.57
	Irregular dental checkup	116 (56.3)	15.19 ± 5.4	
	Regular dental checkup	71 (34.5)	15.49 ± 4.32	
Oral hygiene	Good	89 (43.2)	15.3 ± 4.7	
	Moderate	99 (48.1)	15.55 ± 5.73	0.999
	Poor	18 (8.7)	15.94 ± 7.12	
Bad dental experience	Yes	43 (21.0)	16.95 ± 6.97	0.11
	No	163 (79.0)	15.08 ± 4.87	

## Data Availability

The datasets generated and/or analyzed during the current study are available from the corresponding author upon reasonable request.

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
