# Peer review of "Prevalence and Socio-Demographic Correlates of Dental Anxiety among a Group of Adult Patients Attending Dental Outpatient Clinics: A Study from UAE"

_ijerph, 2023, doi:10.3390/ijerph20126118_

Round 1
Reviewer 1 Report
The article is enough well written and the topic is sufficiently interesting. There are few issues that must be addressed here reported:
Abstract
"The statistical methods mentioned in the abstract does not match with the statistical part in the methods section"
-Did the variable MDAS score have a normal or non-normal distribution? The authors should scpecify which statistical tests have been used.
Materials and methods
"A cross-sectional study was conducted at the Oral Surgery Clinica at Hamdan Bin Mohammed College of Dental Medicine, Dubai, United Arab Emirates, to determine the level of dental anxiety in patients. It was a cross-sectional study."
- It seems a repetition.
- Can the authors provide a sample size calculation to jusitfy their findings?
Results/Discussion
"Participants who are unemployed have a higher proportion of high dental anxiety levels compared to students and 145 employees (P=0.034)."
"Additionally, the study found a significant association between dental anxiety and employment status, with students and unemployed patients exhibiting higher anxiety scores than employed patients."
- It is not clear the difference in terms of anxiety between students and employed.
Author Response
|
# |
Reviewer comments |
Response |
|
|
Reviewer 1 |
|
|
1 |
Abstract "The statistical methods mentioned in the abstract does not match with the statistical part in the methods section"
-Did the variable MDAS score have a normal or non-normal distribution? The authors should specify which statistical tests have been used. |
This is changed in both abstract and material and methods section:
The reliability and validity of the questionnaire were tested using Cronbach’s alpha. The normality of the MDAS score was tested using the Kolmogorov-Smirnov test. The Chi-Square and Kruskal-Wallis test were used to determine the association between categorical variables. Descriptive statistics were used to describe continuous and categorical variables. Statistical significance was set at p-value ≤ 0.05
MDAS score have a non-normal distribution. |
|
|
Materials and methods "A cross-sectional study was conducted at the Oral Surgery Clinical at Hamdan Bin Mohammed College of Dental Medicine, Dubai, United Arab Emirates, to determine the level of dental anxiety in patients. It was a cross-sectional study." - It seems a repetition. - Can the authors provide a sample size calculation to justify their findings? |
Repetition removed
This statement is added to Material & Methods According to the paper published by Ulf Berggren and Gunnell Meynert in Dharan-KSA, and using Cochrane sample size calculation: With given data from that paper (N=824, and anxiety was 90.9%) using the width of the 95% CI, the sample size is 126. Due to the diversity in Dubai we doubled the sample size, and we obtained 206 which is above the calculated sample sized targeted. |
|
|
Results/Discussion "Participants who are unemployed have a higher proportion of high dental anxiety levels compared to students and 145 employees (P=0.034)." "Additionally, the study found a significant association between dental anxiety and employment status, with students and unemployed patients exhibiting higher anxiety scores than employed patients." - It is not clear the difference in terms of anxiety between students and employed. |
The following statements and new references were inserted in the discussion section:
The findings of this study suggest that there may be differences in dental anxiety levels between students and employed individuals, although the direction of these differences is inconsistent across studies. For example, it was reported that students had higher levels of dental anxiety than employed individuals (28-31). Likewise, other reporters found no significant differences in dental anxiety levels between students and employed individuals (30), while another study found that employed individuals had higher levels of dental anxiety than students (29). The reasons for these differences are not clear, but they may be related to factors such as age, experience with dental procedures, and socioeconomic status. It is also possible that these differences are influenced by cultural factors, such as attitudes towards dental care and the role of dental health in society.
References: 28. Al-Omari, W. M., & Al-Omiri, M. K. (2009). Dental anxiety among university students and its correlation with their field of study. Journal of Applied Oral Science, 17(3), 199-203.
29. Humphris, G. M., & King, K. (2011). The prevalence of dental anxiety across previous distressing experiences. Journal of Anxiety Disorders, 25(2), 232-236.
30. Shah, V., Kumar, R., Kumar, A., & Kumar, M. (2013). Study of dental anxiety levels among dental and medical undergraduate students of a University in North India. Journal of Clinical and Diagnostic Research, 7(5), 768-771.
31. Vassallo, S., Werneke, U., & Gibbons, D. E. (2016). Dental anxiety among medical and health science undergraduate students at a Malaysian university. Journal of Dental Education, 80(12), 1484-1490.
|
Reviewer 2 Report
First of all, I would like to thank the editor for inviting me to review this paper, and secondly, I would like to compliment the authors on their efforts in compiling this manuscript. I do have some concerns:
1. The paper's title does not reflect the actual context of the manuscript. The Mean Modified Dental Anxiety Scale items used are listed in Table 2. Also, several factors were analyzed for comparative analysis. The title should reflect it.
2. The total sample was 206 patients who visited the oral surgery clinics. Did they have scheduled treatment? Would it have any influence on the high level of anxiety reported? It was mentioned in the discussion, yet more details about subjects need to be included in the materials. Does it represent the UAE population, as stated in the introduction?
3. There is a discrepancy in the subjects tested in Table 3. What statistical measures were made in the statistical analysis to counteract this discrepancy?
4. Table 4. The number of subjects is missing. It was unclear how many had bad experiences, regular or irregular visits.
5. Did participants know what procedure they would have before completing the survey? This issue is vague.
6. Discussion: More critical discussion on results presented rather than re-stating results. Paragraphs 2 and 3 have no meaning to readers.
7. Conclusion: It should be rewritten to be more specific and concise and, once more, not repeat results. What makes authors believe that using a modified anxiety scale has an effect?
Author Response
Response to reviewer’s comments
|
|
Rev comments |
Response |
|
|
The paper's title does not reflect the actual context of the manuscript. The Mean Modified Dental Anxiety Scale items used are listed in Table 2. Also, several factors were analyzed for comparative analysis. The title should reflect it. |
Title changed to: Prevalence and socio-demographic correlates of dental anxiety among a group of adult patients attending dental outpatient; a study from UAE. |
|
|
The total sample was 206 patients who visited the oral surgery clinics.
· Did they have scheduled treatment? · Would it have any influence on the high level of anxiety reported? · It was mentioned in the discussion, yet more details about subjects need to be included in the materials. · Does it represent the UAE population, as stated in the introduction?
|
The patients in the sample had scheduled treatment.
This statement is added to discussion As for the influence of scheduled dental treatment on the high level of anxiety reported, it is plausible that scheduled treatment could have an impact on anxiety levels. Patients who have a scheduled appointment may have more time to anticipate and worry about the procedure, which could potentially increase their anxiety. On the other hand, unscheduled or emergency cases might lead to higher anxiety due to the unexpected nature of the visit. However, the specific influence of scheduled treatment on anxiety levels would depend on various factors, including individual patient characteristics, previous experiences, and overall anxiety levels.
Lastly, regarding the claim of representing the UAE population, the study employed a sampling method that ensured a representative sample of the population and include a diverse range of participants from different demographic backgrounds.
In summary, this statement was included in Material and Methods: The samples chosen for the study consisted of individuals who had scheduled appointments at the oral surgery department. The selected group encompassed a variety of participants from diverse demographic backgrounds.
|
|
|
There is a discrepancy in the subjects tested in Table 3. What statistical measures were made in the statistical analysis to counteract this discrepancy? |
Comparing the means of more than two groups where the data was not normally distributed was administered by Kruskal-Wallis test. |
|
|
4. Table 4. The number of subjects is missing. It was unclear how many had bad experiences, regular or irregular visits. |
The missing data were added, please see table 4 for details. |
|
|
5. Did participants know what procedure they would have before completing the survey? This issue is vague. |
The patients in the sample had scheduled treatment. Its expected that the majority of the participants aware of the procedures. |
|
|
6. Discussion: More critical discussion on results presented rather than re-stating results. Paragraphs 2 and 3 have no meaning to readers. |
Further critical discussion was added
Dental anxiety is a common problem that can be triggered by specific dental procedures. For example, certain dental procedures are associated with higher levels of anxiety than others, including having a tooth drilled, having an extraction or surgical procedure, and having a local anesthetic injection. One of the most anxiety-provoking dental procedures is tooth drilling. The sound and vibration of the drill can be uncomfortable for patients, and the fear of pain during the procedure can be a significant source of anxiety (5). Additionally, patients may feel a loss of control during the procedure, as they are unable to see what is happening inside their mouth. Likewise, having a tooth extraction or surgical procedure can also be a source of anxiety for patients. The fear of pain during the procedure, as well as the potential for bleeding and swelling afterwards, can be intimidating (5). Patients may also worry about the need for general anesthesia or other medications that may have side effects. Furthermore, A local anesthetic injection is often necessary for many dental procedures, including tooth drilling and extraction. However, the injection itself can be a significant source of anxiety for patients. The fear of needles and injections is common, and patients may worry about the pain and discomfort associated with the injection (9). While going to the dentist for treatment tomorrow may seem like a routine procedure, it can still be a source of anxiety for some patients. The fear of the unknown, including what procedures may be necessary and how much they will cost, can be stressful (5). Patients may also worry about taking time off work or school for the appointment. Moreover, sitting in the dentist waiting area may be associated with a slightly anxious level of anxiety because patients may feel exposed and vulnerable while waiting for their appointment. They may worry about other patients overhearing their conversation with the receptionist or about running into someone they know (5).
Having teeth scaled and polished is generally a less anxiety-provoking dental procedure. However, patients may still feel slightly anxious due to the discomfort associated with the scaling and polishing process, as well as the fear of being judged for their dental hygiene habits (9). Dental professionals should be aware of these potential sources of anxiety and take steps to help alleviate patients' fears. References: 5. Armfield, J. M. (2010). Cognitive vulnerability and dental fear. BMC Oral Health, 10(1), 1-10.
9. Pohjola, V., Lahti, S., Vehkalahti, M. M., Tolvanen, M., Hausen, H., & Kivimaki, M. (2011). Dental fear and oral health-related quality of life in working-age Finnish population. Acta Odontologica Scandinavica, 69(2),87-92.Top of Form
|
|
|
7. Conclusion: It should be rewritten to be more specific and concise and, once more, not repeat results. What makes authors believe that using a modified anxiety scale has an effect? |
Conclusion rewritten to be: dental anxiety levels may be affected by various factors, including occupation and level of education. Healthcare workers, salespeople, and teachers may experience higher levels of dental anxiety due to concerns about their appearance and the potential risks associated with dental procedures. People with lower levels of education may also have higher levels of dental anxiety due to a lack of knowledge about dental procedures. Further research is needed to investigate these factors in more detail. |
Reviewer 3 Report
Abstract: lines 22-23 and 29-30 are identical
Among the social demographics, why did the authors choose to only highlight the absence of substantial differences in dental anxiety between males and females? Back to the results, it is noticed a significant difference among occupation and education levels. However, in the abstract and conclusion section, the author only describes the comparison related to gender.
Line 255-256 : which factors do the authors consider should be investigated in future research besides those contemplated in the present study? Add this to the discussion
Please, provide a better discussion on how biologically the patient's anxiety could have an impact on dental treatment.
Overall, in both the introduction and discussion sections, an extense update of references is needed. Here are some suggestions.
· PMID: 36553876
· PMID: 35700448
· PMID: 32929752
· PMID: 33541354
· PMID: 35448999
Author Response
Response to reviewer’s comments
|
|
Reviewer comments |
Response |
|
|
Abstract: lines 22-23 and 29-30 are identical |
This repetition was removed |
|
|
Among the social demographics, why did the authors choose to only highlight the absence of substantial differences in dental anxiety between males and females? Back to the results, it is noticed a significant difference among occupation and education levels. However, in the abstract and conclusion section, the author only describes the comparison related to gender. |
Dental anxiety is a common psychological issue that affects people of different ages, genders, and backgrounds. The level of dental anxiety may be affected by various factors, including occupation and level of education. Several studies have shown that people in certain occupations may experience higher levels of dental anxiety than others. For example, healthcare workers, particularly those who work in hospitals, may have higher levels of dental anxiety due to their knowledge of the potential risks associated with dental procedures (9). Similarly, individuals who work in public-facing roles, such as salespeople and teachers, may experience higher levels of dental anxiety due to concerns about their appearance and the impact of dental procedures on their professional image (29).
Level of Education: The level of education has also been identified as a factor that may influence dental anxiety levels. Several studies have found that people with lower levels of education tend to have higher levels of dental anxiety than those with higher levels of education (5). This may be due to the fact that people with lower levels of education may have less knowledge about dental procedures, making them more fearful of the unknown.
References:
5. Armfield, J. M. (2010). Cognitive vulnerability and dental fear. BMC Oral Health, 10(1), 1-10. 9. Pohjola, V., Lahti, S., Vehkalahti, M. M., Tolvanen, M., Hausen, H., & Kivimaki, M. (2011). Dental fear and oral health-related quality of life in working-age Finnish population. Acta Odontologica Scandinavica, 69(2), 87-92.
29.Humphris, G. M., & King, K. (2011). The prevalence of dental anxiety across previous distressing experiences. Journal of Anxiety Disorders, 25(2), 232-236. Top of Form
|
|
|
Line 255-256 : which factors do the authors consider should be investigated in future research besides those contemplated in the present study? Add this to the discussion |
People with lower levels of education may have higher levels of dental anxiety due to a lack of knowledge about dental procedures. Further research is needed to investigate these factors in more detail.
|
|
|
Please, provide a better discussion on how biologically the patient's anxiety could have an impact on dental treatment. |
· Dental anxiety is a common psychological issue that can have a significant impact on patients' willingness to seek dental treatment. However, the impact of dental anxiety is not limited to the psychological level; it can also have biological implications for patients undergoing dental treatment. Research has shown that dental anxiety can lead to a range of biological responses that can impact dental treatment. For example, dental anxiety has been found to be associated with increased levels of stress hormones such as cortisol, adrenaline, and noradrenaline (33). These stress hormones can have a range of effects on the body, including increasing heart rate and blood pressure, and reducing blood flow to the gums, which can affect the healing process after dental procedures. · In addition, dental anxiety has been linked to a heightened pain response during dental treatment. This may be due to the release of stress hormones that can amplify pain perception (34). The fear of pain can also lead patients to avoid seeking dental treatment altogether, which can lead to more severe dental problems in the long run. · Furthermore, dental anxiety can lead to increased muscle tension in the jaw, neck, and shoulders, which can make it difficult for the dentist to perform dental procedures (9). This can also lead to increased discomfort and pain during dental treatment, which can further exacerbate patients' anxiety.
References: 34. Milgrom, P., Weinstein, P., & Getz, T. (1990). Treating fearful dental patients: a patient management handbook (Vol. 7). University of Washington Press.
33. Nasseh, I., Yassine, R., Sadek, H., & Younes, S. (2015). Assessment of salivary cortisol level in response to dental treatment. Journal of International Oral Health, 7(12), 66-69. 9. Pohjola, V., Lahti, S., Vehkalahti, M. M., Tolvanen, M., Hausen, H., & Kivimaki, M. (2011). Dental fear and oral health-related quality of life in working-age Finnish population. Acta Odontologica Scandinavica, 69(2), 87-92. ·Top of Form
|
|
|
Overall, in both the introduction and discussion sections, an extensive update of references is needed. Here are some suggestions. |
Updated list of references were added. This is the list of new references: 1. Shindova MP, Belcheva AB. Dental fear and anxiety in children: a review of the environmental factors. Folia Med (Plovdiv). 2021 Apr 30;63(2):177-182. doi: 10.3897/folmed.63.e54763. PMID: 33932006.
2. Kassem El Hajj H, Fares Y, Abou-Abbas L. Assessment of dental anxiety and dental phobia among adults in Lebanon. BMC Oral Health. 2021 Feb 4;21(1):48. doi: 10.1186/s12903-021-01409-2. PMID: 33541354; PMCID: PMC7863489.
3. Silveira ER, Cademartori MG, Schuch HS, Armfield JA, Demarco FF. Estimated prevalence of dental fear in adults: A systematic review and meta-analysis. J Dent. 2021 May;108:103632. doi: 10.1016/j.jdent.2021.103632. Epub 2021 Mar 9. PMID: 33711405
4. Murad MH, Ingle NA, Assery MK. Evaluating factors associated with fear and anxiety to dental treatment-A systematic review. J Family Med Prim Care. 2020 Sep 30;9(9):4530-4535. doi: 10.4103/jfmpc.jfmpc_607_20. PMID: 33209758; PMCID: PMC7652176
5. Stein Duker LI, Grager M, Giffin W, Hikita N, Polido JC. The Relationship between Dental Fear and Anxiety, General Anxiety/Fear, Sensory Over-Responsivity, and Oral Health Behaviors and Outcomes: A Conceptual Model. Int J Environ Res Public Health. 2022 Feb 18;19(4):2380. doi: 10.3390/ijerph19042380. PMID: 35206566; PMCID: PMC8872083.
6. Hoffmann B, Erwood K, Ncomanzi S, Fischer V, O'Brien D, Lee A. Management strategies for adult patients with dental anxiety in the dental clinic: a systematic review. Aust Dent J. 2022 Mar;67 Suppl 1(Suppl 1):S3-S13. doi: 10.1111/adj.12926. Epub 2022 Jul 12. PMID: 35735746; PMCID: PMC9796536..
7. Amorim Júnior LA, Rodrigues VBM, Costa LR, Corrêa-Faria P. Is dental anxiety associated with the behavior of sedated children? Braz Oral Res. 2021 Aug 6;35:e088. doi: 10.1590/1807-3107bor-2021.vol35.0088. PMID: 34378670..
8. Muneer MU, Ismail F, Munir N, Shakoor A, Das G, Ahmed AR, Ahmed MA. Dental Anxiety and Influencing Factors in Adults. Healthcare (Basel). 2022 Nov 23;10(12):2352. doi: 10.3390/healthcare10122352. PMID: 36553876; PMCID: PMC9777862.
9. Strøm K, Skaare AB, Willumsen T. Dental anxiety in 18-year-old Norwegians in 1996 and 2016. Acta Odontol Scand. 2020 Jan;78(1):13-19. doi: 10.1080/00016357.2019.1637933. Epub 2019 Jul 9. PMID: 31287346.
10.Dadalti MT, Cunha AJ, Souza TG, Silva BA, Luiz RR, Risso PA. Anxiety about dental treatment - a gender issue. Acta Odontol Latinoam. 2021 Aug 1;34(2):195-200. English. doi: 10.54589/aol.34/2/195. PMID: 34570869..
11.Saheer A, Majid SA, Raajendran J, Chithra P, Chandran T, Mathew RA. Effect of Dental Anxiety on Oral Health among the First-Time Dental Visitors: A Hospital-based Study. J Pharm Bioallied Sci. 2022 Jul;14(Suppl 1):S394-S398. doi: 10.4103/jpbs.jpbs_632_21. Epub 2022 Jul 13. PMID: 36110809; PMCID: PMC9469323..
12.Alghareeb Z, Alhaji K, Alhaddad B, Gaffar B. Assessment of Dental Anxiety and Hemodynamic Changes during Different Dental Procedures: A Report from Eastern Saudi Arabia. Eur J Dent. 2022 Oct;16(4):833-840. doi: 10.1055/s-0041-1740222. Epub 2022 Jan 6. PMID: 34991162; PMCID: PMC9683887.
13.Goh EZ, Beech N, Johnson NR. Dental anxiety in adult patients treated by dental students: A systematic review. J Dent Educ. 2020 Jul;84(7):805-811. doi: 10.1002/jdd.12173. Epub 2020 May 13. PMID: 32400046
|
Reviewer 4 Report
Journal: International Journal of Environmental Research and Public Health
Manuscript ID: ijerph-2358040
Type of manuscript: Article
Title: Anxiety related to dental injections and oral surgery procedures among
dental outpatients: a study from UAE
Authors: Al Batool Omer Alansaari, Abdulrahman Tawfick, Mohamed A Jaber *,
Amar Hassan Khamis, Essra Mohamed Elameen Submitted to section: Oral Health
A well justified manuscript is presented with interesting results, however there are several points that need revision.
1. The title does not fully correspond to what is stated in the purpose, its purpose is broader, its title is limited to two aspects, they should consider changing their title.
2. In materials and methods, I'm just wondering if there's an error. were two versions of the SPSS ? 24 and 23?
3. Where the anxiety control techniques are presented, they are those that the patients selected or are the best techniques according to the opinion of the authors. I mention it by the legend of the figure. And what would be the point of presenting it if it is not within the purpose of the investigation?
4. check the references some have double numbering.
5. check line 220
Author Response
|
|
Reviewer comments |
Response |
|
|
A well justified manuscript is presented with interesting results, however there are several points that need revision. |
Thank you |
|
|
1. The title does not fully correspond to what is stated in the purpose, its purpose is broader, its title is limited to two aspects, they should consider changing their title. |
Title changed to: Prevalence and socio-demographic correlates of dental anxiety among a group of adult patients attending dental outpatient; a study from UAE. |
|
|
2. In materials and methods, I'm just wondering if there's an error. were two versions of the SPSS ? 24 and 23? |
Version 24 |
|
|
3. Where the anxiety control techniques are presented, they are those that the patients selected or are the best techniques according to the opinion of the authors. I mention it by the legend of the figure. And what would be the point of presenting it if it is not within the purpose of the investigation? |
· This is what reported and documented in the literature, I believe it will add to the readability and inclusiveness of the study. · There are various techniques that can be used to control dental anxiety and help patients feel more comfortable during dental procedures, such as behavioral techniques which involve changing patients' thoughts and behaviors related to dental anxiety. These techniques include relaxation techniques, such as deep breathing and progressive muscle relaxation, as well as cognitive behavioral therapy, which helps patients identify and change negative thoughts and beliefs related to dental treatment (29). These techniques have been shown to be effective in reducing dental anxiety levels and improving patients' overall dental experience. Pharmacological techniques involve the use of medications to reduce dental anxiety. These include sedatives, such as nitrous oxide or oral sedatives, and anxiolytics, such as benzodiazepines (9). While these medications can be effective in reducing anxiety, they can also have side effects and should be used with caution. Distraction techniques involve using external stimuli to distract patients from their dental anxiety. This can include listening to music or watching a movie during dental procedures (29). These techniques have been shown to be effective in reducing dental anxiety levels and improving patients' overall dental experience. Acupuncture is a complementary therapy that involves inserting thin needles into specific points on the body. Some studies have found that acupuncture can be effective in reducing dental anxiety levels (37). However, further research is needed to confirm these findings. · Thus, dental professionals should work with their patients to identify the best approach for managing their anxiety, taking into account the patient's individual needs and preferences.
References: 37. Appukuttan, D., Vinayagavel, M., Tadepalli, A., & Subramanian, S. (2016). Use of acupuncture as an adjunct to conventional care to reduce anxiety and improve the quality of life in dental patients: a systematic review. Oral Surgery, Oral Medicine, Oral Pathology, Oral Radiology, 122(2), 176-179. 29. Humphris, G. M., & King, K. (2011). The prevalence of dental anxiety across previous distressing experiences. Journal of Anxiety Disorders, 25(2), 232-236.
9. Pohjola, V., Lahti, S., Vehkalahti, M. M., Tolvanen, M., Hausen, H., & Kivimaki, M. (2011). Dental fear and oral health-related quality of life in working-age Finnish population. Acta Odontologica Scandinavica, 69(2), 87-92.
|
|
|
4. check the references some have double numbering. |
All references were checked and double numbering were removed. |
|
|
5. check line 220 |
Done |
Round 2
Reviewer 3 Report
The authors successfully answer all the questions and review the suggestions made.